# Gender Bias in Indonesian Courts: Is Perma No. 3 of 2017 the Solution for Gender-Based Violence Cases?

**Rika Saraswati**

School of Law, Faculty of Law and Communication, Soegijapranata Catholic University, Semarang 50234, Indonesia; rikasaraswati@unika.ac.id

**Abstract:** To support women who are dealing with the legal system, especially women victims of gender-based violence, the Indonesian government issued Supreme Court Regulation (Perma) No. 3 of 2017 on Guidelines for Judging Cases of Women in Conflict with the Law. This regulation deals with women as victims, defendants and witnesses, and is used for civil and criminal cases. The Perma appears to attempt to counterbalance existing discriminatory practices in the courts and their processes. This article discusses the effectiveness of the "special treatment" in Supreme Court Regulation (Perma) No. 3 of 2017 on Guidelines for Judging Cases of Women in Conflict with the Law. This Perma seems to provide hope for producing progressive court decisions by contributing to the elimination of discrimination against women in the court process. However, this expectation certainly needs to be reviewed, given that, in their entirety, any such proceedings involve not only judges but also other law enforcement officials, namely the prosecutors. Furthermore, the presence of this Perma is considered by some Indonesian feminists to contradict the Judicial Code of Ethics and Guidelines for Judicial Behaviour ("the Code"). The Code requires judges to be neutral in judging but this Perma demands the opposite. This study is a qualitative study, and the data is obtained through a literature study of research conducted on court decisions and gender-based violence cases involving Indonesian women.

**Keywords:** gender bias; court; equal treatment; special treatment

## 1. Introduction

This article discusses the effectiveness of the Supreme Court Regulation (Perma) No. 3 of 2017 on Guidelines for Judging Cases of Women in Conflict with the Law. The Perma is considered progressive legislation but it is also criticised by Indonesian feminists because of its contradictory Articles. This article also examines the implementation of the Perma in court proceedings that involved a woman who was accused of murdering her baby following its birth.

Indonesian feminists have found that court hearing processes, and the way in which judges question people during the process, undetermined women, primarily as victims (Abdullah et al. 2001) but also where women are alleged. Consequently, court verdicts in cases involving women (and children) as victims of violence are not meeting expectations of fair and equal treatment before the law. Research has shown that a lack of judicial awareness of women's situations and conditions impacts judges' considerations and decision making. Feminists argue that studies of several hearings and court decisions have revealed that the processes also often revictimised women victims, and that again, this was caused by a lack of Indonesian judges' understanding of and their perspective on gender equality issues (Sagala 2005). It is also noted that stereotypes of women persist in judges' perspectives of women. This is particularly so in cases of domestic violence where the stereotype of women as a woman and housewife still holds sway (Nurherwati 2013). Some feminists see this perspective as affecting how victims (and even perpetrators) are treated by judges and the entire justice apparatus.

The stereotype has also been internalised by female victims who may see themselves as ashamed, even guilty or sinful, despite their being victims of violence. This has resulted in women victims being reluctant to report the violence to legal officers, even when they had a strong case. It is well documented that women victims can be positioned as able to lodge complaints that should prompt the prosecution of criminal cases (Abdullah et al. 2001; Komnas Perempuan [National Commission on Violence against Women] 2020); however, they may be reluctant to do so for the social factors which positions their victimhood as shaming not only to themselves but their families, rather than as a cause for legal action and its attendant public exposure of their situation. For those determined to launch a complaint, stereotypes of women as housewife can lead to their not being believed as 'mere women' or the nature of their complaint being minimised to avoid social upheaval.

On the other hand, women as perpetrators also faced difficulties. Like victims, they feel shame, and may believe that they had sinned (transgressed against divine law not just state law). One reason that women may become perpetrators in a domestic violence situation is the patriarchal system within community which creates a power imbalance between male and female, rendering them first as victims but if they retaliate, perpetrators. Women perpetrators feel guilty, tend to blame themselves, and each of the scenarios described women also feeling depressed. As victims, they may be unwilling to share their shame with family (and friends) and reluctant to complain to police. This contributes to a feeling of not being heard, contributing to further distress and depression. Their suffering is no way alleviated in such circumstances. Feelings of shame and guilt may even contribute to an uneasy feeling that they deserve punishment, further silencing them. For those who themselves commit acts of domestic violence, they feel fear when they were positioned as perpetrators of criminal acts, and may believe they need to be punished, even if they were previously victims of violence in the relationship. Women victims (and also perpetrators) are also afraid of community ridicule and 'blaming' for any violence, as they will be seen as unsuccessful wives, unable to fulfil the role of the stereotypical wife and mother as chief peacemaker and key relationship sustainer. They worry, too, about protecting their rights and facing a long trial process, and experience significant trauma because they have to repeat the same story repeatedly during the legal processes which are conducted by police officers, prosecutors, judges and legal advisors (Khusnaeny et al. 2017).

In principle, according to Article 28 D (1) of the Indonesian Constitution 1945, everyone has the right to be treated justly and equally before the law, including being protected from all forms of discrimination in the justice system and when attempting to access justice. Guarantee from all forms of discrimination is necessary because there are difficulties for women in obtaining equality, generally, and particularly, equality before the law and when attempting to obtain justice (Lestari 2016). Permas are created to ensure greater certainty in various aspects of criminal law. Shortcomings identified in court procedure and operations necessitated the creation of Perma No. 3 of 2017 in response to research that had shown that, in many cases, legal officers had not developed satisfactory gender awareness nor demonstrated a high degree of sensitivity when dealing with women involved in domestic violence, generally as victims but sometimes as perpetrators (Aini 2010).

To fulfil the rights of those women who have conflict with law, the Indonesian Supreme Court issued Supreme Court Regulation No. 3 of 2017 concerning Guidelines for Judging Cases of Women in Conflict with the Law (*Perempuan Berhadapan Hukum*/PBH) (Kelompok Kerja Perempuan dan Anak Mahkamah Agung RI dan Masyarakat Pemantau Peradilan Indonesia Fakultas Hukum Universitas Indonesia MaPPI FHUI). Article 1 of the Perma defines women involved in conflict with law as including: women as perpetrators, victims, and witnesses. The purpose of this Perma is to provide guidance for judges so that they can fulfil the principle of gender equality between women and men before the law, especially in court. Moreover, being protected from any form of discrimination is a woman's right which was guaranteed in the Indonesian Constitution 1945 and *Act No. 7 of 1984 on the Ratification of the Convention on the Elimination of all Forms of Discrimination against Women* (CEDAW)). Therefore, it is obligatory for judges (Suprapto 2007) dealing with cases

involving women as victims, perpetrators and witnesses to implement the Articles of Perma No. 3 of 2017. For example, Article 7 of Perma No. 3 of 2017 confirms that:

> During the trial examination, the judge should prevent and/or reprimand the parties, legal advisers, public prosecutors and/or attorneys who behave or make statements that denigrate, blame, intimidate and/or use the experience or sexual background of women in conflict with the law.

Based on this regulation, judges are expected to have adopted a suitable gender perspective and that this perspective would be applied to cases where women are involved as a perpetrator, or victim or as a witness, so that the incidence of discriminatory practices based on gender stereotypes is reduced in court. However, one critic maintains that the regulation is, by its very nature, in breach of the Judicial Code of Ethics and Judicial Code of Conduct ("the Judicial Code of Ethics/Conduct") because judges must apply both the principle of equality and that of special treatment simultaneously, and this would create a dilemma for the judge. The argument is that applying the principle of special treatment for women in conflict with the law will render judges no longer neutral nor able to make fair decisions. Such a failure to remain impartial would thereby permit a judge to be subject to sanctions based on the Judicial Code of Ethics/Conduct (Astuti 2018).

This study is expected to contribute to the literature on concepts about the equal treatment and special treatment of women who are in conflict with the law as perpetrators in the realm of criminal law, following the enactment of Perma No. 3 of 2017. Research on this issue is rare. In contrast, many scholars have undertaken research on women as victims of violence in the criminal justice system (Lestari 2016; Aini 2010; Kelompok Kerja Perempuan dan Anak Mahkamah Agung RI dan Masyarakat Pemantau Peradilan Indonesia Fakultas Hukum Universitas Indonesia  MaPPI FHUI), and women in civil actions such as divorce (whether in the Religious or State Court) or in regard to inheritance (whether adat, Islamic or secular) (Supriadi 2019; Sarini 2018; Maslun 2019; Barlinti 2013). Research is therefore needed to discover the extent of implementation of equal treatment and special treatment in criminal cases where a woman is the alleged perpetrator. This current research comprises a case study that deals with the application of Perma No. 3 of 2017 in court verdict No. 37/Pid.B/2020/PN.Pwd on a case of alleged (maternal) infanticide.

## 2. Material and Methods

This study is a case study undertaken using a qualitative approach. The qualitative study is chosen to determine how the concept of special treatment from Perma No. 3 of 2017 on Guidelines for Judging Cases of Women in Conflict with the Law is implemented and whether it has been accommodated by judges who worked under the regulation within the criminal justice system, and in light of the Judicial Code of Ethics/Conduct, and also to broaden and/or deepen the understanding of how things came to be in terms of women's experiences, especially for the woman's experience in this case study.

Data was obtained through a literature study that centred on a court decision dealing with a gender-based violence case involving an Indonesian woman. The data comprises court verdict No. 37/Pid.B/2020/PN.Pwd on the case of a woman charged with killing her baby after its birth, as well as all documents related to the case, gender-based violence cases and the role of Indonesian judges. The case was chosen because it was about a woman who had experienced sexual violence on the part of her boyfriend and then was accused of killing her baby. The case was recently finalised (June 2020) and proceedings were available through an e-court proceeding online.

All data obtained in this study was processed and analysed using the descriptive analysis method. This method was used to provide a clear and concrete picture of the object discussed qualitatively. After the data analysis was conducted, the data was described by examining the existing problems describing, elaborating, and explaining the problems that relate to the implementation of equal treatment and special treatment.

## 3. Theoretical Framework

There are many feminist theoretical perspectives on equal treatment and special treatment from a maternity and workplace perspective. Some of the theories relevant to this study focus on gender issues in terms of equality and 'being special', and its correlation with a judge's decision making on the basis of Perma No. 3 of 2017, and in an infanticide case in Indonesia. The key elements of the theories used in this paper are discussed below.

### 3.1. What Is a Perma?

A Perma is a regulation issued by the Indonesian Supreme Court to regulate and to guide judges; in the instance of the particular Perma, it is a guideline for the court when it is handling a case where a woman is the perpetrator, victim or witness. The aim of this Perma is to fill a legal vacuum that existed due to there being no adequate legislation as guidance for judges when examining a gender-based case. Thus, this Perma can be said to be a lex specialis towards the Law on Judicial Power and the Judicial Code of Ethics/Conduct.

Judges' duties are regulated in the Judicial Power Act No. 48 of 2009. Article 5 states that judges must investigate, follow, be abreast of the Constitution, and understand legal values and a sense of justice that exists in society. Judges must have integrity, be honest, fair, professional, and experienced in the legal field. Article 4(3) states that judges must obey the Judicial Code of Ethics/Conduct. Article 8 states that every person who is suspected, arrested, detained, prosecuted, or being brought before a court must be presumed innocent until the court convicts that person and such judgement has obtained the permanent force of law. In considering the severity of the punishment, the judge must also pay attention to the good and evil characteristics of the defendant (Article 8(2)).

Judicial duties are also regulated in terms of ethical conduct in a Letter of Decision by the Chairman of the Supreme Court and the Chairman of the Republic of Indonesia Judicial Commission Number 047/KMA/SKB/IV/2009 and 02/SKB/P.KY/IV/2009 on the Judicial Code of Ethics and Judicial Code of Conduct.

One of the principles of the Judicial Code of Ethics and Judicial Code of Conduct is to be fair during court proceedings and in decision making. Being fair means that the treatment of persons should be based on the principle that all people are equal before the law. The most basic demand of justice is to provide equal treatment and opportunities to everyone. Therefore, a judge, as the person carrying out a task or their profession in the field of legal issues has a responsibility to act fairly and not discriminate against people. In general, this regulation gives guidance for the application of the law: (1) Judges are obliged to carry out their legal duties with respect to the presumption of innocence, without expecting any reward; (2) Judges are obliged to be impartial, both inside and outside the court, and still maintain and foster the trust of the people who are seeking justice; (3) Judges are obliged to avoid things that may result in the revocation of their rights to adjudicate a case; (4) Judges are prohibited to impress one of the parties or legal advisor, attorney and witnesses as if one has a special position to influence the judge concerned; (5) Judges are prohibited from showing feelings of like or dislike, partisanship, prejudice, harassment of race, sex, religion, national origin, differences in physical or mental ability, age, or socioeconomic status or on the basis of close relationship with justice seekers or parties involved in the judicial process either through words or actions; (6) Judges are obliged to ask for all parties involved in the trial process to apply the standard of behaviour referred to in point (5); (7) Judges are prohibited from behaving, uttering words or taking actions that give the impression of being partial, prejudiced, threatening, or cornering the parties or their proxies, or witnesses, and the judge must also apply the same standards of behaviour to advocates, prosecutors, court employees or parties, and others who are subject to the direction and supervision of the judge concerned; (8) Judges must provide justice to all parties and not solely intend to punish; (9) Judges are prohibited from ordering/allowing court employees or other parties to influence, direct, or control the proceedings of the trial, thus causing differences in treatment of the parties related to the case.

Based on Article 79 of the Supreme Court Act No. 14 of 1985, the Indonesian Supreme Court can regulate further legal matters that it deems necessary to smooth the processes of the judiciary as long as the matters are not yet regulated in such law, that is a *Peraturan Mahkamah Agung* (Perma) or Supreme Court Regulation. Perma No. 3 of 2017 on Women in Conflict with the Law was issued to protect women's rights when they are in conflict with the law because existing legislation did not sufficiently accommodate the needs and the interests of women victims, particularly as the existing legislation did not embody any gender and human rights' perspectives. For example, in the criminal justice system, the Criminal Procedure Codes do not accommodate the need of victims to be accompanied by a medical or psychological counsellor, nor is there any regulation for the general attorney, when acting as state representation for a victim, to build communication with any victim, their family and counsellors. This omission led to situations where a general attorney treated a victim unfairly by letting the investigator, attorney, judge, or lawyer question in such a way that she felt guilty or made forced (and even false) admissions when put under such pressure. Such situations reflect general gender inequality, and also discrimination against women who are in conflict with the law whose needs are not met in these situations (Sandiata 2018). It is undeniable that the gender bias in legislation can also have a tremendous effect on legal officers in terms of how they do their duties.

This Perma binds judges 'internally' and is used as a guide for judges to apply the principles of respecting human dignity, non-discrimination, gender equality, equality before the law, justice, expedience, and legal certainty (Article 2). The aim of the Perma is to give judges an understanding on how to apply such principles, to identify situations of unequal treatment that result in discrimination against women, and to guarantee women's rights to obtain justice on the basic of equal access (Article 3).

Articles 4 states that judges must consider the principle of gender equality and non-discrimination by identifying the facts that may be before the court, namely: inequality of social status among the litigants, inequality of legal protection that impacts access to justice, discrimination, the psychological effects experienced by a victim, physical and psychological disempowerment of the victim, power relations that contribute to the helplessness of the victims/witnesses, and any history of violence by the perpetrator against the victims/witnesses. A further consideration could be any history of prior violence by a victim of an alleged female perpetrator, particularly if that violence was sustained and serious in nature. It should be recalled that "women in conflict with the law" include perpetrators as well as victims and witnesses.

Furthermore, Article 5 Perma No. 3 of 2017 regulates that during examination of a case of women in conflict with the law, judges may not:

1.  show an attitude or make a derogatory statement, blame and/or intimidate women in conflict with the law;
2.  justify discrimination against women using culture, customary rules,
3.  other traditional practices and gender-biased expert interpretation;
4.  question and/or consider the experiences or the sexual background of the victim as a basis for releasing the perpetrator or reducing the punishment of the perpetrator; and
5.  issue a statement or view that contains gender stereotypes.

The aim of these Supreme Court regulations is to demonstrate that judges have a role to proceed the case fairly and to respect everyone involved in conflict with the law, especially women. It determines explicitly in the Ethics of Conduct and also similarly in Perma No. 3 of 2017, that judges are not allowed to show feelings of like or dislike, partisanship, prejudice, harassment of race, sex, religion, national origin, differences in physical or mental ability, age, or socioeconomic status, or on the basis of closeness, relationships with justice seekers or parties involved in the judicial process either through words or actions, especially for women who have conflict with the law. The Perma is convinced by Indonesian feminists as a legal breakthrough to protect women's rights to access to justice either in the family or criminal justice system.

*3.2. Equal Treatment and Special Treatment*

Equality of treatment is a liberal view that has been primarily concerned with the elimination of laws or social practices which treat women and men in a different way because of their gender, especially in regard to pregnancy and childbearing, and where it affects women's rights in the workplace. The liberal view proposed equal treatment based on two fundamental assumptions. The first is that there is no 'real' difference between the sexes, with the ideal situation being one in which no gender stereotypes exist, as this was generally believed to be the result of normative gender stereotypical socialisation (rather than any innate differences, other than those of lesser importance than the person's common humanity, which should be the foremost consideration). The second assumption is that once the disparate treatment is removed, men and women would achieve equal status through individual freedom of choice and equal competition in the social and economic marketplace (Krieger and Cooney 1993). However, there are dangers and limitations in the liberal view that comes from these two basic assumptions in terms of special conditions such as pregnancy and childbearing because women are particularly affected by pregnancy, childbirth or related medical conditions and special provision needs to be made for them (for example, maternity leave). Otherwise, if they were treated the same as all employees for all employment-related purposes, it could be to the female employees' disadvantage, particularly in the perinatal period, yet special provision may make the female employee less attractive to an employer, especially one that is unaware of the generally lower absentee rate of mothers. Similarly, the concept of equal treatment can pose a danger for judges (as for employers) who possess little understanding on this issue. A lack of understanding can lead to their decision making being less just (than might otherwise have been the case) due to their biases and stereotypes about women.

Special treatment proponents argue that equal treatment is wrong in arguing that women's right to maternity leave is granted on account of their being treated for a medical condition, a situation similar to sick leave taken for physical pain and injury. Special treatment proponents argue that pregnancy occurs because of biological specificity that women possess; therefore, providing maternity leave is a form of preferential treatment, not a form of equal treatment (same treatment) (Finley 1993). Finley argues that both principles (equal treatment and special treatment) do not entirely escape from the male norm. The special treatment approach tends to define the problem by framing it as how the needs of females must be accommodated by the male workplace. By being incorporated into the male workplace, women will be better able to compete with men according to the existing value structures (Finley 1993). The tensions between the equal treatment and special treatment is caused by the male norm in equality theory. The focus of equality theory—its use of the male norm as the 'measuring stick' against which all others are measured and subsidiary to it—makes it well-suited for perpetuating existing distributions of power, as through this power, the (male) norms are the key to the attribution of difference. The male norms are seen as normal and everyone else is 'other'. The consequence predicted that equal treatment has sometimes been used to legitimate discrimination against women rather than to successfully eradicate it.

Finley (1993) quotes Wolgast's approach approvingly:

[W]e need two kinds of rights: equal rights and special rights. Special rights are rights based on human differences, taking them into account so that the ultimate outcome between different individuals can be the same . . .

In her book, *Equality and the Rights of Women*, Elizabeth Wolgast proposed a 'bivalent' view to address debate on the paradigm of sexual equality. Her view differs from the liberal view and rejects the two primary tenets of the liberal feminist view: that sex differences are "illusory", and that equal treatment of the sexes will result in functional equality. She insists that "the differences between men and women are substantial and that sexual equality will result only if society deals with sex differences respectfully and fairly by developing accommodating institutions which permit equality of effect" (Finley 1993). She

acknowledges that the conditions of the sexes are asymmetrical or heterogeneous, at least in some respect. Therefore, this condition sets out to devise a conception of equality that takes this asymmetry into account.

Wolgast asserts the work of two types of rights: 'equal' rights and 'special' rights through her illustration:

> Within our society, every individual is deemed to have an 'equal' right of access to public building. That this right is an 'equal right' means that with respect to that right, any or a person is interchangeable with any other. The right adheres to every individual. But, the effect of no ramp is a denial of the equal right. In such a circumstance, equality is effectuated only if the disabled person is granted a 'special' right to a ramp.

Wolgast's illustration demonstrates that the failure to provide a 'special' right to members of a disadvantaged group because they deviate from the norms have denied them an 'equal' right to which they are entitled (Finley 1993, pp. 168–69). Wolgast's model acknowledges the fact of heterogeneity with the theoretical construct of interchangeability underlying traditional egalitarian thinking. She acknowledges the fact that institutions and policies, such as employment policies, building access and legal policies were designed in accordance with a normative standard to which some groups within society do not conform. By affording those individuals a special right, it means their difference is accommodated, and institutions are modified so that the principle of interchangeability is restored: "So long as the building has a ramp, the walker and a wheelchair user can be substituted one for the other," she maintains. Therefore, women do not have to be proven homogeneous with men in order to gain admission to the 'society of equals' because the bivalent view provides for changes in societal institutions to accommodate differences (Finley 1993, p. 170).

The illustration given by Wolgast is appropriate to the case of Perma No. 3 of 2017 versus the Act of Judicial Power and the Judicial Code of Ethics/Conduct, so the critique that judges will face a dilemma because of the need to accommodate two non-equivalent or mutually contradictory content of the principles of equal treatment and special treatment, can be diminished. The next question is whether judges understand this policy as a special treatment and have applied this when they handle court proceedings, as well as how does the implementation of this treatment affect women in conflict with the law to obtain justice. Llyod (1970) argues that there is a gap between formal and substantial justice because justice requires equality of treatment in accordance with the classifications laid down by the rules, but it says nothing about how people should or should not be treated.

Theo Huijbers argues that the role of the court is to resolve cases where citizens have different perceptions on an issue. The court must resolve the case on the basis of the principle of justice. For this reason, the court is required to treat the parties equally (Huijbers 1990). It can be said that legal certainty can be attained if the Criminal Procedure Code is applied to cases involving women. In such cases, the judges' decision making is based on the indictment by the Public Prosecutor (the details of which are compiled through an investigation report prepared by the Police) and the evidence that is proven during trial examination. In addition, during the proceedings, the judge must adhere to the principle of the presumption of innocence. Judges must not assume the defendant is guilty before all stages of the examination have been completed. Thus, judges must be neutral and apply the principle of equal treatment to women who are in conflict with the law. However, when the Code of Criminal Procedure exhibits gender bias (as mentioned above), it will affect those women because judges will apply it as if it were just. Therefore, special treatment or special arrangements in favour of the poorer or the vulnerable in the community (such as women) are needed to enable them to seek justice on an equal footing with those who possess natural, social, or economic advantages.

### 3.3. The Causes of Infanticide in Indonesia

In Indonesian culture of having a baby outside of marriage is seen as shameful and immoral conduct (Isnawan 2018). Society generally regards giving birth to a baby

conceived in a relationship with a man outside of marriage embarrassing, a terrible event and a despicable occurrence. When a woman is involved in such conduct, she will try and keep her pregnancy a secret for as long as possible, sometimes, even until the baby is delivered.

The commission of the crime of infanticide by the biological mother has been usually caused by a number of factors. First is the fear of being caught giving birth to a child outside of marriage, which has occurred because the mother has been involved in an illicit relationship, which means having sexual intercourse outside of marriage (which includes either with her consent or without it, such as due to sexual violence) (Isnawan 2018; Erika et al. 2019).

A second factor can be a lack of knowledge and support. She may not initially recognise that she is pregnant. She may lack basic knowledge regarding conception and pregnancy and may interpret an interruption in the menstrual cycle as within her normal range (depending on her own personal experience), and movement or pains in the abdomen as indigestion; or she may be reluctant to acknowledge her pregnancy. The woman who is pregnant and gives birth to a baby outside of marriage, especially a woman for whom it is her first pregnancy, does not have any experience of being pregnant, nor does she have any experience in delivering a baby. The entire process brings her feelings of anxiety and fear because of her uncertain life in terms of her pregnancy and her future baby. These feelings are worse if the woman faces pregnancy and enters labour (and all situations that may occur during that time) without support or a companion, either a partner or family member. Furthermore, if the pregnant woman is a high school student, she will attempt to keep her pregnancy hidden because she can be expelled from the school and forced to discontinue her education.

The third factor that affects such women is the social and legal punishment norm. In the legal aspect, she has no marriage bond that gives her and the child status in society. Instead, the birth of a child out of wedlock creates shame for her because the baby does not have a father through a legal marriage and the child may have no father acknowledged on their birth certificate, giving him or her the dubious status of illegitimacy. Besides that, the woman who became pregnant outside of marriage feels ashamed because it seems that everyone knows about her actions that violate legal, religious and social norms. When the norms of a society are not followed and obeyed, sanctions follow. These sanctions can be in the form of ridicule, ostracism by society generally, as well as family and friends. A woman may be sent away by her family. The basis for the fear of being known to give birth to an illegitimate baby is rooted in the societally-agreed reprehensible nature of such births. The act of pregnancy outside of marriage has long been regarded negatively by society, and many find it difficult to accept a pregnant woman outside of marriage (Istiana 2020). This violation of societal, family, religious and legal norms, and the response of society to it, creates multiple burdens of guilt. In some instances, this can lead a woman to access abortion (Uyun and Saputra 2012; Firdausita 2017). Others, however, feel constrained not to do so, seeing this as a worse option or perhaps as an option not available to them. Based on this description, it is undeniable that Indonesian society (not merely the biological mother) also has a significant role through its norms in the occurrence of infanticide (Firdausita 2017; Brennan 2018).

Fourthly, the psychological and emotional state of the mother comprises the last factor. The time at which the crime is undertaken is associated with a highly emotional mental state from the mother with emotions such as shame, fear, hatred, and confusion, together with the pain of childbirth, all mixed together. Therefore, at the time of delivery of the baby, the mother is not in a calm, conscious, and measured mental state. The feelings of fear and shame during pregnancy have grown over time and have emerged an impulse inside her to kill what she sees as the source of her intense, overwhelming distress. This is regarded as a mental disorder that may arise in new mothers, particularly those without any support or with pre-existing mental health difficulties, or who have themselves been abused or neglected. Such negative feelings may arise and then explode into aggression or

other psychotic behaviour, and the impulse to take her baby's life, a thought that may not have originally been her intention (therefore, unplanned), while for others, it is a planned act of desperation (Isnawan 2018; Kartono 1998).

The feeling of shame and guilt does not diminish the act of the biological mother who has killed her baby. Judges who handle such a case (even if it has mitigating reasons) tend to exhibit a gender bias by making a judgement that incorporates an assumption of a mother's greater responsibility and instinct to care for a child. For example, "The defendant's act was inhuman because she killed her own child or her flesh and blood". As a biological mother of this baby, she should care for and protect her child, and care for it with great affection. A mother is assumed to instinctively be full of compassion, gentleness and patience; whatever is sacrificed for the sake of the child. Mothers have an instinct to protect their children and their children's rights so that they can live, grow, develop and participate optimally according to their dignity and human dignity, as well as protecting children from violence and discrimination. Therefore, a mother who commits infanticide must not have such characteristics (Isnawan 2018).

Judges, by concentrating on the act alone, sometimes do not consider the psychological and sociological impacts that the biological mother has suffered due to the unwanted pregnancy. Elsewhere in the world, it is recognised that infanticide may occur due to a specific mental condition called 'postpartum psychosis' which may occur in mothers up to about 12 months or even longer after a baby's birth. In such circumstances, expert witnesses are called to testify to the mental state of the mother. Although access to recent research may be lacking, access to expert witnesses uncertain, and understanding and implementation in terms of sentencing may be inconsistent in some areas, an attempt is made to distinguish clearly between 'premeditated murder' and infanticide attributed to post-partum psychosis or an exacerbation of pre-existing mental illness, or coercion ( Spinelli 2004). In Indonesia, the judge must not only have intellectual ability, morality and integrity, but also the courage to be released from the 'words' of legal norms because the content of the law can actually become a barrier to the woman who wishes to seek justice (Dhofiyah 2019; Hoesein 2016). Due to gender bias in courts, Indonesian feminists have encouraged the Indonesian government to improve the legal system in dealing with court proceedings and judges' perspective. This demand has been responded to in Perma No. 3 of 2017. Indonesian feminists welcomed its provision as a step forward for the world of the justice system in Indonesia.

Below is the Case Study, where the extent of implementation of equal treatment and special treatment in criminal cases, where a woman is the alleged perpetrator, is examined since the introduction of Perma No. 3 of 2017.

*3.4. The Case: Court Verdict No. 37/Pid.B/2020/PN.Pwd*

A woman was accused of having delivered a baby and killing her baby at birth. The case initially went to the State Court in Purwodadi district. The defendant was K, a 21-year-old Muslim woman and employee who lived in Penawangan village. She was interviewed and the case proceeded on the basis of an accusation that she had allegedly acted in a manner that caused the infant's death, having done so because of her fear of being revealed as having delivered a baby, the alleged murder having been committed at the time the baby was born or shortly thereafter. She was threatened with a charge of "murder by design" that is, premeditated murder—having intentionally caused the death of a person, not manslaughter (having unintentionally caused the death of a person).

The origin of the case dated back to when a friend introduced her to a man, RZ, in the beginning of May 2019. She said that she did not like him actually, but the man continued to contact her through WhatsApp messages, in which he said that he wanted to get to know her further and asked her to "hang out". One day she was picked up by RZ, and they went out and wandered around Purwodadi city. After that, she was taken to a small hotel (She did not know its name because she was unaware of the hotel). According to K, the man checked into the hotel, but then grabbed her, taking her into the hotel room. She maintained

that she had tried to run away but that she could not do that because RZ had hidden the room key. RZ had sexual intercourse with her without her consent. This occurred only once. When he had finished, he took her back to the home of her grandparents' house with whom she was living. A week later, RZ could not be contacted and RZ had even blocked her phone number.

In June and July 2019, K did not menstruate. She bought a pregnancy test at a chemist warehouse and then tested her urine. The result was positive. She did not tell anybody about her pregnancy. She attempted to keep her pregnancy hidden because she was afraid of being expelled from the house by her family. Moreover, she did not have a permanent job at the moment. Then, on 9 January 2020 at 3.00 a.m., she felt extreme pain in her stomach due to being in labour. She went to the bathroom and then, in a semi-squatting position, delivered the baby. The baby was delivered with its placenta; however, K observed that the baby did not cry. Then, she cleaned the baby and wrapped it with a plastic bag and put it into a basket. She took the baby out from the plastic bag and then threw the baby into a fish pond and placed the placenta in water waste pipe.

Two days later, the body that had been thrown into the fish pond rose in the water and it was found by her uncle and grandfather. Based on what they had found, they reported to the head of the village and a police officer. Police then investigated the scene with the help of a midwife from the village community health centre. Based on their investigation, K was the only woman in that house who had the symptoms of a woman who had delivered a baby. Therefore, the police detained her on suspicion of killing the baby under Articles 341 and 342 (which, if she were found guilty, would have resulted in prison sentences of up seven or nine years, respectively). The prosecutor, in this case, the general attorney, asked for her (if convicted) to serve two years in prison under Article 341 and 342 of the Criminal Code.

The Articles cited by the prosecutor were:

341—the mother who . . . with deliberate intent takes the life of her child at or soon after its birth, shall, being guilty of infant-manslaughter, be punished by a maximum imprisonment of seven years.

342—the mother who . . . with deliberate intent [and premeditation] takes the life of her child at or soon after its birth, shall, being guilty of infanticide, be punished by a maximum imprisonment of nine years.

### 3.5. Judges' Decision

The Panel of Judges rejected the charge laid by the general attorney on the basis of Articles 341 and 342 of the Criminal Code because they found that the charge and Articles applied were irrelevant to the facts revealed in the court proceedings. Therefore, the panel of judges made their own deliberation and made their own consideration, and argued that the defendant's act fell under Article 181 of the Criminal Code rather than Articles 341 and 342 of the Criminal Code. Article 181 states:

Any person who buries, hides, take away or removes a corpse with the intention of concealing his death or birth, shall be punished by a maximum imprisonment of nine months or a maximum fine of four thousand five hundred rupiah.

The considerations of the panel of judges were: First, the baby's body was only examined externally and without an autopsy being conducted. Based on the results of the visum et repertum, the exact cause of death cannot be determined. Second, testimony from the experts as witnesses convinced the judges that the cause of the death could not be known exactly because no autopsy had been undertaken. Judges were not convinced about when and why the baby died because according to the expert witness, the baby was probably already dead before it was born because the contractions in the process of delivering the baby was faster than that for a baby who is born alive, so the baby slid out suddenly and then hit the bathroom floor.

Third, judges determined that the defendant's conduct from labour to throwing the baby's body into the fish pond showed her panic, fear and the effort taken to hide her pregnancy and the birth of the baby. Based on this fact in the court hearing, the judges were convinced that the defendant must be declared legally guilty and was convincingly proven to have committed the crime of hiding her baby after the baby was born.

Fourth, the judges rejected the defence offered by the defendant's legal advisor who requested the defendant be not legally proven guilty and not be convincingly found to have committed a criminal act as accused by the general attorney as public prosecutor under charges derived from Articles 341 and 342 of the Criminal Code, and subsequently released the defendant from the Public Prosecutor's indictment. The judges argued that the defendant's actions were not proven to have violated the Articles named in the charge, but declared that is was proven that the defendant's actions did constitute a criminal act as they violated Article 181 of the Criminal Code.

Fifth, judges had considerations which were burdensome and relieved the defendant. According to judges, the circumstances were aggravating as the defendant's actions did not reflect a love of a mother for her child. On the other hand, the mitigating circumstances were: (1) The defendant has never been convicted, she admitted her deed, and was honest and polite during the trial proceedings; (2) The defendant showed remorse, felt sorry and promised that she would not repeat her actions; (3) The defendant was a victim from a perpetrator who did not want to take any responsibility for a pregnancy that he initiated; (4) The village community still accepted the defendant because the defendant was well known as a good member of the community and had never committed any crimes before.

Based on the evidence witnesses provided in the court hearing and proceedings, the Judges rejected the charge that had been laid by the general attorney and decided to sentence her to only nine months imprisonment on the basis of Article 181 of the Criminal Code rather than Articles 341 and 342 of the Criminal Code.

## 4. Discussion—Equality before the Law v. Special Treatment

Based on the examination during the court proceedings, it can be seen that judges had applied the equal treatment and the special treatment provision at the same time in this case. However, how far the special treatment is applied and how far it can influence the equal treatment will be discussed further.

Judges applied the principle of equality before the law as—equal treatment—by treating the woman who was accused of killing her baby almost immediately following its birth as a defendant and by assessing that their task was to determine whether she is guilty or not. Equal treatment was also undertaken by judges when they based their decision on the application of the legal norm, that is Article 181 of the Penal Code. Although the judges acknowledged that the defendant was a victim of sexual violence, they held firmly to their decision based on that article. If judges had considered the situation of the defendant, as a victim of sexual violence, as 'special', then this fact is supposed to be used and treated by judges as a 'special treatment'—they would have been able to release the defendant from charges and punishment.

There is another 'special' situation that can be used and considered by judges in such circumstances; that is the letter from the head of the village community. In this instance, the letter stated that the defendant was a good person and would be welcomed after being released from prison. It is acknowledged that the defendant was a good woman, and she was a victim of a man who did not take any responsibility. The case history of the defendant revealed that her experience of being a victim of sexual violence had led her to infanticide; and the letter from the head of the village community demonstrated a 'special' situation and condition that should be considered by judges. However, these non-legal facts seemed not to significantly influence the findings and cannot change the legal norm; therefore, consideration was made on the basis of Article 181 of the Penal Code. This situation has demonstrated that equal treatment remains a dominant factor, while the application of 'special' treatment will be influenced by the willingness of judges to determine whether

special treatment is applicable or not. This situation has led to discrimination against women and gender inequality.

Another example of the implementation of equal treatment and special treatment in this case is with an examination of Perma No. 3 of 2017 in terms of the judges' obligation to carry out their legal duties with respect to the presumption of innocence and prohibition against behaving, uttering words or taking actions that contribute to an impression of being partial and prejudiced. Such ethical conduct is expected to be implemented as a part of 'equal' treatment for everyone before the law. Equal treatment requires judges to treat the defendant as not guilty on the basis of the presumption of innocence until all stages of the examination are finished. Thus, judges must be neutral, namely, applying the principle of equal treatment to women who are in conflict with the law.

However, the researcher found several statements from the judges that indirectly blame or intimidate women in dealing with the law, and this contradicted with Article 5 of Perma No. 3 2017 which states that during examination of the women in conflict with the law, judges may not show such an attitude or makes derogatory statements or blame and/or intimidate such women. The following examples illustrate how the judges' statements may superficially appear "normal" but, in fact, indirectly blame the defendant for her situation:

> After all, everything is finished, yes, everything has happened. Nothing can be fixed except regret, and improve yourself in the future, right? It will be your worst experience so that is not easy; and what kind of strangers want to take you, and you must have the courage to refuse because you don't have the courage and assertiveness and don't be afraid of something that may not happen ...

The judge's statement that seemed to blame the defendant indirectly by saying " ... it is not easy to be carried away by a stranger ... " and "you must have the courage to refuse because you do not have courage and assertiveness ... ". The statement can be interpreted as offering advice to the defendant that she must be careful in the future, not be easily carried away by strangers and must have a firm attitude and dare to refuse. The advice demonstrates a belief that the defendant may remain very vulnerable to violence because she lacked courage or assertiveness and did not exercise her power to refuse the man's requests. This attributed blame for the situation to the victim (later, perpetrator) to her inability to refuse to accompany a man rather than to the man who apparently abandoned her after allegedly having sexually assaulted her (with a resulting pregnancy perhaps due to his failure to use a condom). It can be hazarded that many Indonesian women have faced similar situations because of an imbalance of power that is established socially, religiously and culturally.

The judge unwittingly also made a statement which intimidated the defendant by referring to the case as a "killing", even though the case was still in the process of being heard. The judge interrupted and reprimanded the defendant's legal adviser who was questioning witnesses (from the National Women's Commission). The judge argued that the questions were irrelevant to the case:

> For legal advisors, the focus is on cases because the case is infanticide, not violence against women. Do not focus on violence against women but on cases of dumping babies [into the pond] or (baby) killing, please focus on this case, not go everywhere ... (Saraswati et al. 2020)

This statement certainly contradicts the presumption of innocence. By referring to this case as one where similar cases would be of "dumping babies or (baby) killing", the judge indirectly indicates a belief that the defendant had indeed killed the baby, while evidence presented in the examination of witnesses and expert witnesses revealed there were still doubts about the exact cause of the baby's death.

Furthermore, the word "killer" was also repeated by the judge when hearing the defendant's testimony, even appearing to attribute the death to a birthing accident during

delivery—but the judge still referred to the newborn as a victim—and more of K's inability to act initially (perhaps when approached by the man):

> Actually this is not your destiny, because you do not have the will and dare to act so that it ends up like this. So that your child must become a victim. I'm not saying you are a killer or anything, because it wasn't handled properly during childbirth.

The judge also made a statement that the defendant had committed a sin. This was revealed when the judge asked the gender of the baby who had been thrown into the pool: "Oh, girl. You just need to pray for your child, hopefully your sins will be forgiven because whatever happened is your sins". The judge's statement was a form of blaming the defendant and using religious belief to further attribute blame. The issue of whether the outcome was a result of actions that the judge clearly believed were sin is actually irrelevant in this case. It is not the judge's domain to judge or determine whether a human act is sinful or not. The domain of the judge at trial is to determine whether a person violates legal provisions or not.

The judge's statement, whether it was made consciously or unconsciously, is basically a form of revictimisation carried out by a legal officer at the trial. The issue of revictimisation has already been raised by the criminal expert witness who stated that the defendant actually had experienced revictimisation as a result of male seduction and coercion to have sex without her consent (Court verdict: 21–22). Then, when she became pregnant, the man who had impregnated her left. The defendant carried the baby to term and gave birth on her own. She had had no assistance at any time during the pregnancy and now was responsible for everything herself. The defendant's attitude in not revealing her pregnancy to anyone but instead hiding until she gave birth is the result of stigmatisation of sex outside of marriage. The community regards it as a family disgrace, views the women (and their children) as not good, sinful, and so on. Women victims of sexual violence are also stigmatised in the community. Victim blaming and victim shaming are typical responses that lead to under-reporting and perpetrators going unprosecuted and unpunished while the victims bear the consequences of injury (physical and emotional) and sometimes, pregnancy. In this instance, the defendant was also actually a victim of sexual violence. The stereotyping of such victims is well-documented (Krahé 2016, p. 671). As a result, the defendant was confused, panic-stricken and fearful when she delivered the baby. If there is no stigma from society towards a woman who is pregnant outside of wedlock, the defendant's sad experience would not have happened. Meanwhile, the man who impregnated the defendant was not touched by the law (Krahé 2016, p. 671). This discrimination against women has contributed to women being socioeconomically disadvantaged and viewed as culturally inferior to men (Luhulima 2006, p. 87).

The description above has demonstrated that during the court proceeding, judges have implemented Articles in Perma No. 3 of 2017 as a 'special' treatment, but gender bias remains in the judges' response or questions. The tension between equal and special treatment in the court proceeding is a direct result of the male norm—the norm that has power. With equal treatment using the male norm, the implementation of equal treatment and special treatment undeniably will be undertaken from their power and their perspective. Judges are the party who have the power and authority to use it, and even have authority to make the attribution of difference. Based on their power and perspective, they will see themselves (and their experience) as normal and everyone else as the 'other' or undesirable other (Finley 1993). This theoretical framework can be used to explain the decisions made by judges that intend to give punishment rather than to release the defendant, even though the charges were not proven because of lack of evidence.

Judges have used their male norms through their own experiences and perspectives to examine the defendant's experience, therefore, through their 'optics', the defendant's experience indeed is categorised as not normal. As the psychologist witness said:

> According to us, as normal people, the action of the defendant is considered irresponsible (not normal), but for her (or people who are 'loose') it was a respon-

sible (or normal) action because according to her (or them), it was considered as the right solution and normal. So, the decisions taken are influenced by the situation, condition and education of the defendant. (Saraswati et al. 2020)

The situation, condition and education are the main key for whether women are treated as equal (or not). The defendant's experience demonstrated that she had been treated unfairly by a man who could scarcely even been characterised as her boyfriend, and as a result, when living in the "male's world", suffered patriarchy's stigmas as a bad woman, having premarital sex, becoming pregnant outside of marriage, and having a baby out of wedlock. All the stigmas had made her afraid to disclose her relationship and pregnancy to her family. Finley insists that " . . . the standards of legal equality have been determined by those who have had the power—in this case is judges—to define or to imprint their view of what is real, important, and normal on others" (Finley 1993, p. 198). Based on the case, the implementation of equal treatment and special treatment, especially for women in conflict with the law, still put women in a position that is not yet profitable. As Williams (1993, p. 151) said, the court (and judges) somehow are not the place to seek such important changes. Women still experience violence, and as victims of violence, women do not receive protection to get equal treatment in order to obtain justice. Women, as victims of sexual violence, remain in the shadow and fear of society's negative stigmatisation of them. The stigma has created discrimination and prevents women from getting equal treatment and special treatment at the same time. Changing the perspective of society towards women who become pregnant outside of marriage is also needed. Meyer and Oberman in Brennan (2018, pp. 4–5) said that "Infanticide is not a random unpredictable crime. Instead it is deeply imbedded in and is a reflection of the societies in which it occurs.".

This situation happens not only in Indonesia but could happen also in other states if the domination of the 'male's world' remains existing in court and the criminal justice system (Smith and Skinner 2012; Douglas 2016; Temkin et al. 2018). Therefore, special circumstances need to be considered and taken into account in determining whether matters come to trial, as is allowed in Indonesia, and in sentencing where such matters do come to trial. The role of expert witnesses, and ease of access to them, could be further explored.

## 5. Conclusions

Perma No. 3 of 2017 is a tool that the Indonesian Supreme Court has given judges as guidance to deal with women who are in conflict with the law. The Perma provides Articles to treat women who are in conflict with the law in a specific way by considering their gender difference and power relation. Judges are expected to have a neutral gender perspective and that perspective is applied to cases in which women are involved as perpetrator, victim or witness, so that discriminatory practices based on gender stereotypes can be reduced in court. On the other hand, judges must implement the principle of equality before the law and examine the case fairly for everyone regardless of their gender identity. Based on the case study in this research, judges have implemented Articles in Perma No. 3 of 2017 as a 'special' treatment during the court hearing, but gender bias remains expressed in the judges' responses or questions. The tension between equal and special treatment is a direct result of the male norm being the norm that has power. Judges are the party who have the power and the authority to use it; they even have authority to attribute differences as to whether the defendant's experience is real or unreal, important or unimportant, and normal or not normal. Based on the case examined here, the implementation of equal treatment and special treatment, especially for women in conflict with the law, has not yet resulted in consistently genuine equal and appropriate special treatment, as the male norm through the equal treatment remains more dominant than the special treatment. Special circumstances, such as non-legal factors, need to be considered and taken into account in determining the court's decision.

**Funding:** This research was funded by Faculty of Law and Communication, Soegijapranata Catholic University.

**Institutional Review Board Statement:** Ethical review and approval were waived for this study, due to the research focused on the court proceeding and secondary data.

**Informed Consent Statement:** Informed consent was obtained from LRC-KJHAM, a non-government organization, that was acting as a legal advisor for the defendant in the case.

**Data Availability Statement:** Not applicable.

**Conflicts of Interest:** The author declares no conflict of interest.

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
