# Peer review of "Gender Bias in Indonesian Courts: Is Perma No. 3 of 2017 the Solution for Gender-Based Violence Cases?"

_laws, 2017_

Round 1
Reviewer 1 Report
The goal of this paper is to discuss the effectiveness of the Indonesian Supreme Court Regulation (Perma) no. 3 of 2017 relating to Guidelines for judging cases of women in conflict with the law (including offenders, victims and witnesses).
According to the author, the guidelines provided by this regulation introduce a "special treatment" for women in order to contribute to eliminate discrimination against women in the court process.
The paper discusses the concepts of "equal treatment" and "special treatment" (according to Wolgast's theory) and examines a case brought before a court in 2020 to verify whether discriminatory practices based on gender stereotypes still remain despite Perma 3/2017.
The overall impression is that in contrast with what is declared by the Author in lines 128-130, not a detailed, systematic and comprehensive picture of all matters is proposing but just an initial discussion of the issues arising from the Perma. The first part of the discussion of the case study (lines 439-497) is not convincing (Are judges not truly independent if they accept, for instance, the request of the defendant's lawyer? what is the author definition of a "very normative and positivist" perspective?).
The presentation must be improved: there are a lot of typos (e.g., in lines:53, 58, 60, 70, 117, 204, 278, 282, 295 etc.); the structure of paragraph "Theoretical Framework" is not clear (see lines 144 and 239); paragraph named "Results" should have a different title; the numbered list in lines 223-229 is not correct; the diagrams do not support the comprehension of the text.
Author Response
I have uploaded my response and the revision of manuscript

Reviewer 2 Report
This is an interesting article on an area we don't read much about in English, so for this reason alone it would be of interest to your readers. It's very well written, though it will need a lot of copy-editing - small grammatical and spelling issues, not general style. It's a case study, set in well-explained legal context, but it would have benefited from some social context too - most readers won't know much about women's position in Indonesia and especially this woman's circumstances and ideas about women's place and so on. Something on the feminist influence on law in Indonesia would also be interesting. There was a good discussion of the central issue of equality versus special treatment approaches but most of the sources seemed quite old - there is surely much more recent feminist analysis of this in possibly similar contexts. And some suggestions for ways forward would help; we have the analysis and critique, but where do we go from here? The argument could usefully be broadened out to make it more significant, either beyond Indonesia or beyond this one case study.
Author Response
I have uploaded my response and the revision of manuscript.

Round 2
Reviewer 1 Report
The issues in my comments have been addressed in the new version of the paper.